# Using Dextran Instead of Egg Yolk in Extender for Cryopreservation of Spermatozoa of Dogs of Different Ages

**DOI:** 10.3390/ani12243480

**Published:** 2022-12-09

**Authors:** Taisiia Yurchuk, Olena Pavlovich, Maryna Petrushko

**Affiliations:** Department of Cryobiology of Reproductive System, Institute for Problems of Cryobiology and Cryomedicine, National Academy of Sciences of Ukraine, Pereyaslavska 23, 61016 Kharkiv, Ukraine

**Keywords:** cryopreservation, spermatozoa, Chinese Crested, dog, canine, motility, DNA fragmentation

## Abstract

**Simple Summary:**

In this study we compare the impact of cryopreservation with dextran and egg yolk on motility, morphology, and DNA integrity of spermatozoa of dogs of different ages (Chinese Crested breed). We found that the concentration, total number, and motility of fresh spermatozoa decreased, whereas the damage of the DNA increased in dogs older than 7 years. The cryopreservation of spermatozoa using extenders with egg yolk or dextran led to a decrease in these parameters in the oldest age group in an equal manner. However, taking into account the possibility of standardizing the composition of the freezing media and excluding foreign proteins from it, the use of dextran for freezing dog spermatozoa is preferable. The cryopreservation of dog spermatozoa, especially of the Chinese Crested breed, should be carried out in the young and middle aged dogs due to the age-related decrease of the cryotolerance of the cells.

**Abstract:**

Egg yolk is a very common supplement of extenders aimed to protect sperm from cryoinjury, but due to their biological risks and difficulties with media standardization, there is a search for alternative. In addition, sperm cryoresistance can be affected by the initial decrease of their functional characteristics caused by age. The aim of this work was to evaluate the efficiency of using dextran (molecular weight 500 kDa) in the extenders instead of egg yolk for the cryopreservation of spermatozoa of dogs (Chinese Crested breed) of different ages. The obtained ejaculates were divided into three groups depending on the animal’s age: 1–3, 4–6 and 7–10 years old. Sperm was cryopreserved by using 7% glycerol and 20% egg yolk, or 20% dextran. The cryoresistance of spermatozoa of the oldest age category was dramatically decreased, which was manifested in their morphology, motility, and DNA fragmentation rate. There were no differences between the cryoprotectant effect of the dextran-based extender on spermatozoa and the egg yolk-based extender in all age categories of dogs. However, given the benefits of dextran-containing media, its use for the cryopreservation of canine spermatozoa has potential benefits that need to be confirmed by sperm fertilization outcomes.

## 1. Introduction

The artificial insemination of animals is a highly effective method for improving the pedigree qualities of animals as well as enhancing their productivity. It is not widespread in breeding work with dogs, although in recent years, cases of artificial insemination using cryopreserved dog semen have become more frequent in many countries in the world for medical and/or breeding management reasons [1]. Another benefit of cryopreservation is the ability to use cryobanked sperm from outstanding sires to support genetic diversity [2]. In addition, cryopreservation protocols for dog semen cryopreservation may be used to preserve the semen from threatened wild Canid species [3]. However, cryopreservation induces a decrease in morphological and functional characteristics of spermatozoa due to oxidative stress, which can also cause lipid peroxidation and DNA fragmentation [4,5,6].

The degree of cryopreservation damage can be determined by the initial morphological and functional states of spermatozoa. Taking into account that the morphology and motility of spermatozoa decline as the dog’s age increases, it can adversely affect the sperm’s cryotolerance [7,8]. The degree of cryopreservation damage also depends on cooling-thawing rates and freezing medium composition determining their protective properties [9,10]. Cryoprotectant media for many animal species, including dogs, most often include glycerol and egg yolk [11,12,13]. Egg yolk amino acids are widely known to have cryoprotectant and antioxidant effects on dog spermatozoa, however, it is difficult to standardize by compounds composition, which varies with bird diets [14]. Moreover, the use of chicken yolk always carries the risk of cross-contamination of samples with various infectious agents and has possibility to change spermatozoa membrane properties while acting with foreign egg yolk proteins [15]. In this regard, the search for substances that can replace the yolk in the composition of the cryoprotectant medium is relevant. The presence of polysaccharides in the cryoprotectant media lowers the crystallization temperature, which facilitates the deep cooling of cells [16]. In this regard, the use of dextran as a non-penetrating cryoprotectant with glycerol is likely to be able to perform the protective role instead of egg yolk. It was noted that dextran forms a layer around membrane phospholipids due to the formation of bonds between the hydrogen of phosphate groups and the -OH group of dextran [17,18]. In addition, the ability to bind dextran through weak hydrogen bonds and Van der Waals interaction was shown [19]. Thanks to these connections between dextran and the sperm surface, the formation of large ice crystals is reduced, and, thus, mechanical damage to the cells is reduced [20].

The aim of this work was to evaluate the cryotolerance (morphology, motility, and DNA integrity) of spermatozoa of dogs of different ages after freezing with dextran- (molecular weight 500 kDa) or egg yolk-based media.

## 2. Materials and Methods

The procedures involving dogs were approved by the Ethical Committee for Animal Experimentation of the Institute for Problems of Cryobiology and Cryomedicine of the National Academy of Sciences of Ukraine. (ECAE-IPCC; #15.2018). Written informed consent was obtained from the owners before their animals participated in the study.

### 2.1. Semen Collection and Analysis

The ejaculates of 9 male Chinese Crested dogs aged between 1 and 10 years old (3 dogs in each age group) from a private breading centre were used in the research. Weekly, the semen was collected manually (three times for each dog) in the presence of a female in the period of oestrus. Immediately after that, the second fraction from each ejaculate was placed in a water bath at 37 °C and transported to the laboratory in 1 h and then analyzed.

Concentrations of spermatozoa were determined by cytometric method in a hemocytometer. Sperm viability was assessed in smears stained with eosin-nigrosin (Magapor, Spain) under a light microscope with a magnification of ×400. The morphological violations index describes the percentages of head, midpiece, and tail abnormalities. The percentage of total motile spermatozoa (TMOT), progressive motile spermatozoa (PMOT), sperm velocity parameters curvilinear velocity (VCL), average path velocity (VAP), straight line velocity (VSL), amplitude of lateral head displacement (ALH), and straightness (STR) were determined by computer-assisted sperm analysis (CASA; SpermVision, Minitube), as described previously [21]. Dogs whose semen had total motile sperm higher than 80% and normal morphology, and sperm counts higher than 200 × 10^6^ sperm cells per 1 mL were considered fertile.

### 2.2. Sperm Preparation and Cryopreservation

Ejaculates obtained from dogs were divided into groups depending on the age of the animals: group 1—age 1–3 years, group 2—4–6 years, group 3—7–10 years (Figure 1). Then, the samples were diluted with Tris-citric acid-fructose (TCF) extender in a ratio of 1:1 and centrifuged at 700× *g* for 5 min (room temperature). TCF composition: 249 mM Trizma base (Sigma-Aldrich, St. Louis, MO, USA), 80 mM citric acid, 69 mM fructose, supplemented with 0.1g penicillin, 0.1g streptomycin sulfate. Sperm pellets were re-suspended in TCF at a concentration of 200 × 10^6^ spermatozoa/mL. Canine sperm freezing was performed as described by Rodenas et al. with our modifications [22]. Diluted spermatozoa in a tube were plunged into a 250 mL glass beaker containing 200 mL of water at 4 °C for 1 h. Then, samples were diluted slowly by adding an equal volume of freezing extender 1 (14% glycerol, 40% of egg yolk in TCF) or freezing extender 2 (14% glycerol, 40% dextran (Sigma-Aldrich, USA) in TCF) that was pre-cooled up to 4 °C, resulting in a final concentration of 100 × 10^6^ spermatozoa/mL. After 30 min of equilibration with the freezing extender, the sperm was packed into 0.25 mL plastic straws (Minitube, Tiefenbach, Germany) that were placed horizontally on a rack 4 cm above the surface of liquid nitrogen (LN_2_) for 15 min. Then, they were plunged into the LN_2_ and kept in it for at least 1 week before being thawed for evaluation. The straws were thawed in a water bath at 38 °C for 30 s. The content of each straw was diluted in a stepwise manner with TCF, and after centrifugation the pellets were diluted with TCF. After that, we assessed motility, viability, and DNA fragmentation rate.

### 2.3. Sperm DNA Fragmentation Assessment

Determination of the DNA fragmentation rate was carried out using the Halosperm kit (Halotech, Spain) according to the protocol specified by the manufacturer. The principle of determining the specified indicator is based on the SCD (sperm chromatin dispersion) method—determination of sperm chromatin dispersion [23]. Sperm were immobilized in an agarose gel on a glass slide, treated with an acid solution for DNA denaturation, and then membranes and proteins were lysed with a buffer. Next, after fixation in an ethanol solution, the samples were stained with a solution of eosin and thiazine to visualize dispersed DNA loops. Sperm with fragmented DNA had very small or no dispersion halos, whereas sperm with low levels of fragmentation released DNA loops that form large halos. The preparations were visualized under an “AmScope B120C” light microscope (AmScope, Irvine, CA, USA), and the 200 spermatozoa were assessed per one smear by two researchers independently.

### 2.4. Statistical Analysis

Statistical comparisons were made using GraphPad Prism software (version 9.3.1; Graphpad Software Inc., San Diego, CA, USA). Comparisons of the semen volume and sperm concentration with age groups were made using the Kruskal–Wallis test with multiple comparisons to test for significance of mean differences. Comparisons of progressive motility, morphologically abnormal spermatozoa, and DNA fragmentation rates with time (pre-freeze and post-thaw) as within age groups were made using two-way ANOVA multiple comparisons. There were considered to be mean differences when there was a *p*-value < 0.05. Results in figures are depicted as scatterplots as individual ejaculate and median values.

## 3. Results

Volume of the sperm-rich ejaculate fraction increased with the dogs’ age until 6–7 years and was the biggest in dogs in the oldest group (7–10 years old; *p* < 0.05) (Table 1). Sperm concentration decreased with age and differed significantly between the study groups; *p* < 0.05. There were no significant differences in total sperm count per ejaculate between groups 1 and 2, whereas the lowest value was in group 3 (*p* < 0.05).

The number of morphological abnormal spermatozoa increased with age and were statistically different between all groups (*p* < 0.05 and *p* < 0.0001, respectively for group 1 vs. group 2 and group 2 vs. group 3) (Figure 2). Cryopreservation in extender with either egg yolk or dextran resulted in an increase in the number of sperm with abnormal morphology in group 3 up to 51.6 ± 10.2. A slight suppressive effect on the morphological characteristics of spermatozoa was observed after cryopreservation in age group 2 (25.3 ± 5.1) compared to group 1 (14.3 ± 3.1; *p* < 0.05). However, there were no significant differences of this group 2 indicator between egg yolk and dextran extenders (*p* ˃ 0.05).

Abnormalities of the head, midpiece, and tail of spermatozoa as well as multiple abnormalities increased with the age of the animals (Table 2). There were more spermatozoa detected with tail pathology in group 2, multiple abnormalities in group 3, and fewer midpiece abnormalities in group 3 after cryopreservation in the egg yolk extender, compared with similar age groups after using the dextran-based extender. However, it should be noted that these differences in the number of certain morphological abnormalities forms did not lead to significant changes in the total number of abnormal sperm morphologies between the used extenders.

The percentages of total motile spermatozoa decreased with age and differed significantly for groups 1 vs. 2 and 2 vs. 3 (*p* < 0.05 and *p* < 0.0001, respectively) (Figure 3). The most dramatic changes in cell motility after cryopreservation were observed in group 3 (33.9 ± 8.4) (*p* < 0.0001) compared to other age groups. In addition, cryopreservation with both types of extenders led to a decrease in the studied indicator in relation to fresh cells (*p* < 0.0001).

The percentage of progressive motile spermatozoa decreased in dogs of group 2 and declined dramatically after cryopreservation of spermatozoa of group 3 regardless of the use of an extender (Table 3). VCL, VAP, and VSL decreased in fresh sperm of dogs older than 7 years and in cryopreserved sperm of dogs older than 3 years regardless of the use of an extender. The parameters of ALH and STR did not change significantly in all study groups.

The analysis of the DNA integrity also showed that the number of cells with fragmented DNA increases with age (Figure 4). Thus, in fresh spermatozoa of dogs older than 7 years, the level of DNA fragmentation increased significantly (21.2 ± 2.3) in relation to young individuals of 1–3 years of age (7.7 ± 2.5; *p* < 0.0001). After cryopreservation using both extenders, a decrease in DNA integrity was observed in all age groups, but the highest index was in group 3 (39.4 ± 7.8; *p* < 0.0001). It should also be noted that significant differences in the level of the studied indicator when using two extenders were not observed in samples of all age animals (*p* ˃ 0.05).

## 4. Discussion

### 4.1. Age-Related Cryotolerance of Canin Spermatozoa

It has been shown in many species of animals and some breeds of dogs that the cryotolerance of spermatozoa decreases with age [8,24,25]. The results of our study have shown that Chinese Crested dogs experience changes with age that affect the total volume of ejaculate and the number of spermatozoa, along with their morphological characteristics and motility and the integrity of the DNA structure. These changes already begin in middle age (4–6 years) and progress the most after 7 years. These data are confirmed by the results of other authors, who showed the appearance of subfertile groups starting from middle age and the absence of such in young dogs of the Labrador Retriever breed [7]. Another recently published study noted a decrease in normal sperm morphology, membrane integrity, and cell motility in dogs older than 10 years of age [8]. However, there was no distribution by breed in that study, therefore, it is possible that certain discrepancies with our data are due to the peculiarities of the onset of age changes in representatives of the Chinese Crested breed in our case and in Labrador Retrievers, which were reported by other authors [7]. An interesting feature of the Chinese Crested breed is its exterior, namely the presence of fur only in certain minor areas of the body. As is known, this occurs due to a mutation in the FOXI3 gene, which causes the development of ectodermal dysplasia, and individuals with a mutant FOXI3 gene demonstrate hairlessness and are heterozygotes for this mutation [26]. However, powderpuff representatives of this breed are completely covered with fur and are not a carrier of this mutation. In our study, all dogs had a hairless exterior, that is, they had one dominant mutant allele of the FOXI3 gene in their genotype. FOXI3 belongs to forkhead box (FOX) family transcription factors and plays essential roles in development, tissue homeostasis, and diseases [27]. It is shown that knockout of another gene of this FOXO3 family induces premature aging [28]. Therefore, it is likely that individuals who carry a mutant allele of the FOXI3 gene are also prone to earlier aging, which ultimately affects the spermatogenesis of animals. Age characteristics, in turn, lead to a change in the cryosensitivity of cells to cryopreservation factors that cause their damage [8]. Therefore, special attention is paid to additives to the cryopreservation medium affecting sperm freezing outcomes.

### 4.2. Using Dextran Instead of Egg Yolk as a Component of Cryoprotectant Media

It is believed that egg yolk as a component of extender hasit’s the ability to reduce the osmotic stress of spermatozoa at various stages of cryopreservation [29]. It realizes its effects due to its participation in the regulation of sperm volume during osmotic fluctuations and changes in the properties of the phospholipids of their membrane [30,31]. It is known that the addition of various di- and polysaccharides to the composition of cryoprotectant media containing penetrating cryoprotectants also helps to regulate the volume of cells during osmotic changes that occur at various stages of cryopreservation and increase the glass transition temperature of the freezing extender and the difference in heat capacity associated with the glass transition [32,33]. Therefore, we hypothesized that the use of dextran in a canine sperm freezing extender may be equivalent to an egg yolk extender. Since dextran is a natural polysaccharide with a molecular weight of up to 20,000 kDa and exhibits the properties of an extracellular cryoprotectant, it has successfully been used for freezing mesenchymal stem cells [34] and blood cells [35]. The use of dextran for the cryopreservation of epididymal spermatozoa of goat [36], turkey [20], and more recently for the preservation of rabbit spermatozoa [37] and boar [17] has been reported. Dextran can protect membrane phospholipids from the negative impact of reactive oxygen species arising in the process of cryopreservation, and thus, like egg yolks, perform an antioxidant function as part of the extender [38]. Taking into account that cryopreservation induces reactive oxygen species production that can also impact on the DNA integrity, we evaluated this index after using dextran- and egg yolk-based cryoprotectant media. Our studies showed any significant differences in the number of spermatozoa with intact DNA between the extenders used. The greatest decrease in these indicators was observed after cryopreservation in the older age group, which indicates a decrease in cryoresistance to the action of cryopreservation factors, regardless of the composition of the medium used. Thus, the broken DNA integrity of gametes and their motility caused by age-related changes may still have latent damage, which is especially clearly manifested after cryopreservation. The same effect of two types of extenders on the motility of spermatozoa of dogs of different ages was observed after thawing. Therefore, the replacement of the egg yolk volume in the extender with an equivalent volume of dextran (m.w. 500 kDa) does not lead to an increase in spermatozoa damage that can impact motility and DNA structure.

## 5. Conclusions

It should be noted that while consulting the owners of particular Chinese Crested dogs, it was found that the quality of fresh and frozen spermatozoa can decrease starting from middle age and thereby affect the results of both natural and artificial insemination. The spermatozoa cryotolerance after cryopreservation either with dextran- or egg yolk-based media showed no difference in all age categories. However, given the benefits of cryoprotective media containing dextran, its use for cryopreservation of canine spermatozoa has potential benefits that need to be confirmed by sperm fertilization outcomes.

## Figures and Tables

**Figure 1 animals-12-03480-f001:**
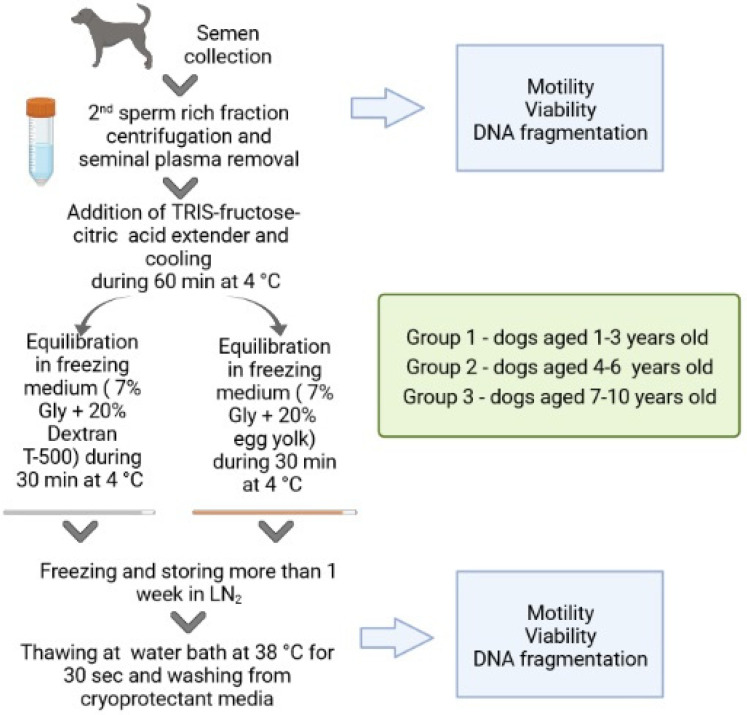
Experimental design for semen collection, data processing, and analysis.

**Figure 2 animals-12-03480-f002:**
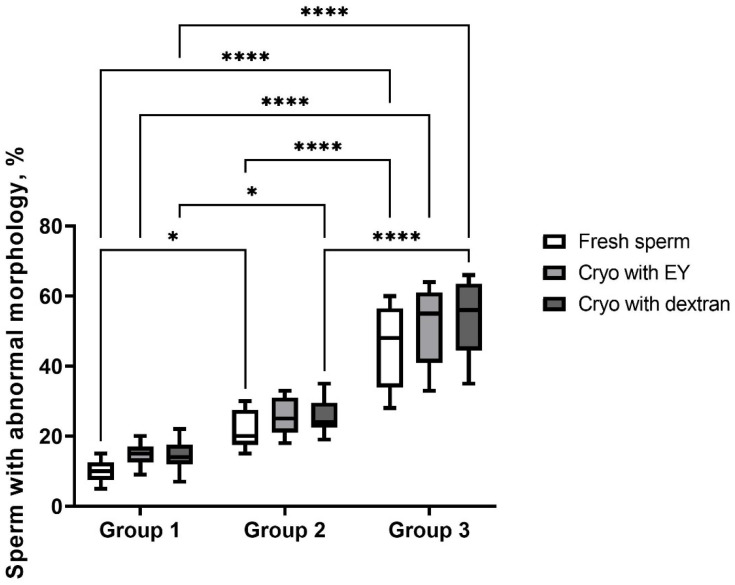
Morphological characteristics of spermatozoa of dogs of different ages after cryopreservation with two different extenders. *—*p* < 0.05, ****—*p* < 0.0001.

**Figure 3 animals-12-03480-f003:**
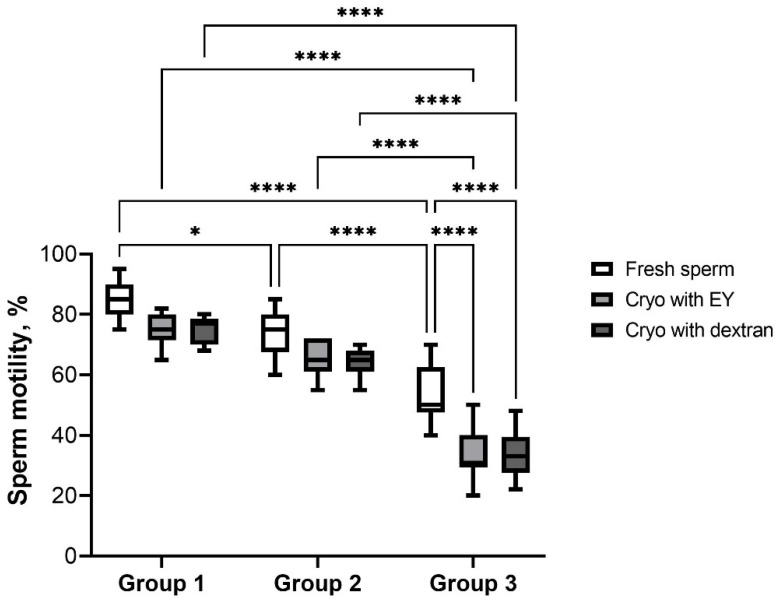
Total motility of spermatozoa of dogs of different ages after cryopreservation with two different extenders. *—*p* < 0.05, ****—*p* < 0.0001.

**Figure 4 animals-12-03480-f004:**
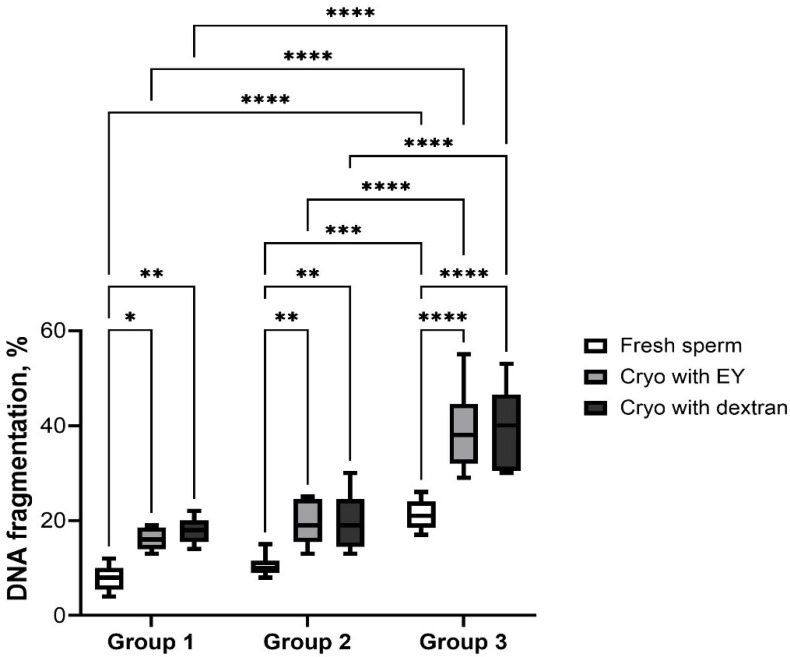
DNA fragmentation rate of dogs of different ages after cryopreservation with two different extenders. *—*p* < 0.05, **—*p* < 0.005, ***—*p* < 0.0005, ****—*p* < 0.0001.

**Table 1 animals-12-03480-t001:** Volume of the sperm-rich fraction of ejaculate, sperm concentration, and total sperm count in ejaculates collected from dogs of different age groups (values are means ± SD).

Age Group	Group 1	Group 2	Group 3
Semen volume, mL	0.99 ± 0.23 ^a^	1.66 ± 0.27 ^a^	2.87 ± 0.88 ^b^
Sperm concentration, ×10^6^ cell/mL	523.3 ± 120.4 ^a^	274.4 ± 21.86 ^b^	73.89 ± 32.86 ^c^
Total sperm count, ×10^6^ cell	498.4 ± 78.45 ^a^	453.0 ± 69.93 ^a^	201.4 ± 77.75 ^b^

Within a row, different superscripts indicate significant differences (*p* < 0.05).

**Table 2 animals-12-03480-t002:** Percentage of spermatozoa with various morphological abnormalities depending on the age of the dog and the type of extender.

	Group	Morphology Abnormalities			
		Head, %	Midpiece, %	Tail, %	Multiple Abnormalities, %
Fresh	1	1.7 ± 0.4 ^a^	1.8 ± 0.9 ^a^	1.5 ± 0.7 ^a^	4.9 ± 0.5 ^a^
	2	3.7 ± 0.9 ^b^	3.3 ± 0.6 ^b^	3.4 ± 0.5 ^b^	11.11 ± 1.5 ^b^
	3	9.1 ± 1.1 ^c^	8.9 ± 0.7 ^c^	6.3 ± 0.7 ^c^	21.3 ± 2.1 ^cd^
Cryo with EY	1	2.3 ± 0.3 ^a^	2.5 ± 0.5 ^ab^	3.8 ± 0.4 ^b^	5.9 ± 1.0 ^a^
	2	3.9 ± 0.7 ^b^	4.2 ± 0.3 ^b^	7.4 ± 0.9 ^c^	10.1 ± 0.9 ^b^
	3	10.0 ± 1.2 ^c^	9.1 ± 0.5 ^c^	9.3 ± 1.0 ^c^	22.9 ± 1.9 ^d^
Cryo with dextran	1	2.1 ± 0.4 ^a^	2.3 ± 0.4 ^a^	3.7 ± 0.2 ^b^	6.3 ± 0.7 ^a^
	2	4.3 ± 0.4 ^b^	4.1 ± 0.7 ^b^	5.3 ± 0.4 ^b^	12.2 ± 1.0 ^b^
	3	12.2 ±1.1 ^c^	12.2 ± 1.2 ^d^	11.8 ± 1.1 ^c^	17.6 ± 1.5 ^c^

Within a column, different superscripts indicate significant differences (*p* < 0.05).

**Table 3 animals-12-03480-t003:** Parameters of sperm motility measured by CASA.

	Group	Motility Parameters						
		TMOT, %	PMOT, %	VCL, µm/s	VAP, µm/s	VSL, µm/s	ALH, µm	STR
Fresh	1	84.4 ± 6.3 ^a^	67.9 ± 6.1 ^a^	175.3 ± 7.0 ^a^	140.5 ± 2.7 ^a^	112.8 ± 3.2 ^a^	2.2 ± 0.3	82.2 ± 0.8
	2	73.3 ± 7.9 ^b^	55.3 ± 6.1 ^b^	171.8 ± 8.1 ^a^	136.6 ± 3.7 ^a^	99.7 ± 5.6 ^a^	2.1 ± 0.2	81.5 ± 1.2
	3	53.9 ± 9.6 ^c^	25.7 ± 3.9 ^d^	155.8 ± 6.9 ^b^	112.9 ±3.2 ^b^	91 ± 5.1 ^b^	2.0 ± 0.3	80.1 ± 0.9
Cryo with EY	1	75.1 ± 5.3 ^b^	42.1 ± 3.7 ^c^	164.3 ± 5.7 ^a^	129.8 ± 3.2 ^a^	108.4 ± 4.1 ^a^	2.1 ± 0.4	79.8 ± 1.1
	2	65.6 ± 6.2 ^c^	34.1 ± 5.0 ^d^	155.9 ± 7.2 ^b^	110.7 ± 3.5 ^b^	94.6 ± 3.2 ^b^	2.0 ± 0.5	78.9 ± 1.5
	3	33.9 ± 8.9 ^d^	18.6 ± 3.4 ^e^	148 ± 6.6 ^b^	103 ± 4.7 ^b^	87.9 ± 3.6 ^b^	1.9 ± 0.2	77.8 ± 0.9
Cryo with dextrann	1	74.8 ± 4.5 ^b^	40.6 ± 4.5 ^c^	166.1 ±7.2 ^a^	127.6 ± 3.2 ^a^	106.9 ± 4.9 ^a^	2.1 ± 0.2	80.1 ± 0.9
	2	64.2 ± 5.0 ^c^	37.0 ± 3.6 ^d^	151.2 ± 7.2 ^b^	115.6 ± 4.3 ^b^	92.7 ± 3.9 ^b^	1.9 ± 0.4	77.9 ± 0.7
	3	33.3 ± 8.03 ^d^	17.8 ± 4.3 ^e^	146.7 ± 5.3 ^b^	109 ± 3.2 ^b^	88.2 ± 3.5 ^b^	1.8 ± 0.3	79.3 ± 1.1

Within a column, different superscripts indicate significant differences (*p* < 0.05).

## Data Availability

The data presented in this study are available on request from the corresponding author. The data that support the findings of this study are available in this article.

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
