# Peer review of "Using Dextran Instead of Egg Yolk in Extender for Cryopreservation of Spermatozoa of Dogs of Different Ages"

_animals, 2022, doi:10.3390/ani12243480_

Round 1

Reviewer 1 Report

L3: ...Egg Yolk In Extender

L8: In this study, ...

L22: ...

 of dog spermatozoa (Chinese Crested breed)...

L76: the symbol of temperature to be supercript.

L86: the number 6 to be supercritp. Please, change it in all points of the manuscript.

L99: How was the concentration of dextran selected? Did the authors perform any pre-trial about it?

L162-4: After cryopreservation using both extenders, an increase in DNA fragmentation was observed in all age groups, but the highest index was in group 3 (39.4±7.8; p<0.0001).

L189-200: Please, minimize the extended description about the FOXI3 gene or may include this information (in brief again) in the introduction part.

L219-27: Please, include these sentences in the introduction in the point where the dextran is described. A brief description about components of cryopreservation extenders related to membrane protection from mechanical, osmotic, oxidative stress could be written to substitute the above mentioned removed sentences.

Author Response

Response to Reviewer 1 Comments

Point 1. L3: ...Egg Yolk In Extender

 L8: In this study, ...

L22: ... of dog spermatozoa (Chinese Crested breed)...

L76: the symbol of temperature to be supercript.

L86: the number 6 to be supercritp. Please, change it in all points of the manuscript.

Response 1: Thank you very much for these comments, we have corrected all these mistakes

Point 2. L99: How was the concentration of dextran selected? Did the authors perform any pre-trial about it?

Response 2: We evaluated several concentrations of dextran (10%, 20%, 30% and 40%) and used the best 20%.

Point 3. L162-4: After cryopreservation using both extenders, an increase in DNA fragmentation was observed in all age groups, but the highest index was in group 3 (39.4±7.8; p<0.0001).

Response 3: Thank you very much, we have corrected this mistake

Point 4. L189-200: Please, minimize the extended description about the FOXI3 gene or may include this information (in brief again) in the introduction part.

 Response 4: We agree that the description of the gene may be slightly extended, but if we transfer this to the introduction, then it would be logical to present the results of the study of the relationship between the FOXI3 gene mutation and aging and reduced fertility. However, in this paper, we did not investigate this and we only shared our assumption, so this information in the introduction may disappoint the reader.

Point 5. L219-27: Please, include these sentences in the introduction in the point where the dextran is described. A brief description about components of cryopreservation extenders related to membrane protection from mechanical, osmotic, oxidative stress could be written to substitute the above mentioned removed sentences.

Response 5: We are grateful for this comment; we followed it and make the appropriate changes.

Reviewer 2 Report

Title:

The title is not appropriate. It should be revised to clearly indicate that no difference was found between dextran and egg yolk.

Abstract:

L29-30: The author may want to argue that dextran is better than egg yolk in terms of quality stability and risk of microbial contamination, but the principle of cryopreservation of sperm is to ensure and demonstrate fertility after insemination. It is necessary to weaken the tone of the argument.

M & M:

L81-84: It was stated that VCL and VAP were measured in Materials and Methods, but I do not believe the results are listed.

L89-90: Male dogs were divided into three groups by age, but it is not clear how many males were tested in each group. This is very important information.

Results:

Since fresh sperm quality deteriorates with age, it is not surprising that sperm quality after cryopreservation in Group 3 is the poorest.

The abnormal morphology, motility, and DNA fragmentation should be compared to see how much each parameter increases or decreases after freezing and thawing in both egg yolk and dextran supplementation.

Conclusions:

L246-248: Such claims should be made only after the results of artificial insemination are obtained.

Author Response

Response to Reviewer 2 Comments

Point 1Title:

The title is not appropriate. It should be revised to clearly indicate that no difference was found between dextran and egg yolk.

 Response 1. Thank you for this suggestion. We have changed the title: Using Dextran Instead Of Egg Yolk In Extender For Cryopreservation Of Spermatozoa Of Dogs Of Different Ages

Point 2. Abstract:

L29-30: The author may want to argue that dextran is better than egg yolk in terms of quality stability and risk of microbial contamination, but the principle of cryopreservation of sperm is to ensure and demonstrate fertility after insemination. It is necessary to weaken the tone of the argument.

 Response 2.  You are wright that we cannot conclude about fertilization ability of the sperm cryopreserved with dextran because we did not check it yet but in line 22- 30 (‘However, taking into account the advantages of dextran based cryoprotectant media, their use for dog spermatozoa cryopreservation is promising) we want only to underline the some benefit of using dextran instead of egg yolk.

M & M:

Point 3. L81-84: It was stated that VCL and VAP were measured in Materials and Methods, but I do not believe the results are listed.

 Response 3. We thank you for you direct and honest comment.

Below we provide detailed results of sperm motility. We did not want to overload the paper and include only result of total motility. If you think that we should include it we will add.

Motility parameters

TMOT, %

PMOT, %

VCL, µm/s

VAP, µm/s

VSL, µm/s

ALH, µm

STR

Fresh

Group 1

84.4 ± 6.3 a

67.9 ± 6.1 a

175.3 ± 7.0a

140.5 ± 2.7a

112.8 ± 3.2 a

2.2 ± 0.3

82.2 ± 0.8

Group 2

73.3 ± 7.9 b

55.3 ± 6.1 b

171.8 ± 8.1 a

136.6 ± 3.7 a

99.7 ± 5.6 a

2.1 ± 0.2

81.5 ± 1.2

Group 3

53.9 ± 9.6c

25.7 ± 3.9d

155.8 ± 6.9 b

112.9 ±3.2 b

91 ± 5.1 b

2.0 ± 0.3

80.1 ± 0,9

Cryo with EY

Group 1

75.1 ± 5.3 b

42.1 ± 3.7c

164.3 ± 5.7a

129.8 ± 3.2 a

108.4 ± 4.1 a

2.1 ± 0.4

79.8 ± 1.1

Group 2

65.6 ± 6.2c

34.1 ± 5.0d

155.9 ± 7.2 b

110.7 ± 3.5 b

94.6 ± 3.2b

2.0 ± 0.5

78.9 ± 1.5

Group 3

33.9 ± 8.9d

18.6 ± 3.4e

148 ± 6.6b

103 ± 4.7 b

87.9 ± 3.6 b

1.9 ± 0.2

77.8 ± 0.9

Cryo with dextrann

Group 1

74.8 ± 4.5 b

40.6 ± 4.5c

166.1 ±7.2a

127.6 ± 3.2 a

106.9 ±4.9a

2.1 ± 0.2

80.1 ± 0.9

Group 2

64.2 ± 5.0c

37.0 ± 3.6d

151.2 ± 7.2 b

115.6 ± 4.3 b

92.7 ± 3.9b

1.9 ± 0.4

77.9 ±0.7

Group 3

33.3 ± 8.03d

17.8 ± 4.3e

146.7 ± 5.3 b

109 ± 3.2b

88.2 ± 3.5b

1.8 ± 0.3

79.3 ±1.1

Within a column, different superscripts indicate significant differences (p < 0.05)

Point 4. L89-90: Male dogs were divided into three groups by age, but it is not clear how many males were tested in each group. This is very important information.

 Response 4. Thank you for this comment. We had 9 dogs and there were 3 dogs in each age group. We collected sperm 3 times from each dog. We added this information to the material and methods.

Point 5. Results:

Since fresh sperm quality deteriorates with age, it is not surprising that sperm quality after cryopreservation in Group 3 is the poorest.

Response 5. Our results have shown when the decline of the sperm quality occur in dogs and particularly in Chinese Crested Bread. This information is important for ones who want to preserve semen of the dog of this bread.

Point 6. The abnormal morphology, motility, and DNA fragmentation should be compared to see how much each parameter increases or decreases after freezing and thawing in both egg yolk and dextran supplementation.

Response 6. We compare fresh, and cryopreserved sperm in both extenders between all age groups in each histogram of mentioned indexes.

Point 7. Conclusions:

L246-248: Such claims should be made only after the results of artificial insemination are obtained.

Response 7. Thank you for this suggestion. We can change this sentence: “However, given the benefits of cryoprotective media containing dextran, their use for cryopreservation of canine spermatozoa has potential benefits that need to be confirmed by sperm fertilization outcomes.”

Reviewer 3 Report

The influence of age on the quality of canine sperm has been investigated in many studies and therefore the topic of this work is not innovative. In turn, the use of dextran for cryopreservation instead of yolk can be a good technological solution in the cryopreservation of canine semen, as shown by the results of this work. Therefore, undertaking such research may be interesting and innovative.

My main comments on the paper submitted for review refer mainly to the methodology and presentation of the results. I include them below:

Line 72 - please provide information on how many animals were included in the study in each age group. Is n = 9 for each age group?

Line 79 - there is no information on how the sperm morphology was assessed, what dye, what method did the authors use?

The analysis of the results is incomplete and needs to be completed. In the description of the methodology, the authors state that they analyzed the parameters of sperm motility using the CASA system, and therefore they should present these results (PMOT, VCL, VAP, VSL, ALH and STR values) for both extenders, taking into account different age groups.

Similarly, in the analysis of sperm morphology, it would be worth showing changes within different sperm structures, i.e. head, midpiece and tails, which were assessed and listed in the methodology description (line 80).

Other comments:

Line 86 - should be corrected ‘106 ’by ‘106’. This revision should be made throughout the text where this error occurs.

Line 76 - the sign of degrees Celsius  ‘oC’ should be corrected and elsewhere in the text.

Line 85 - should be ‘morphology’, not ‘porphology’.

The line 96 - expression ‘diluted spermatozoa’ should be replaced with e.g. ‘diluted samples of sperm’

Line 103 - expression ‘LN2’ should first provide the full name, and then use the abbreviation.

In Figure 1, there are linguistic errors that need to be corrected (it should be ‘sperm’, not ‘serm’; ‘seminal, not ‘semenal’; ‘thawing’, not ‘twaing’).

Line 141 - in the caption Table 1 is better to put ‘in the same row’ instead of ‘in the same line’.

Figure 3 and Figure 4 should be included in the results, not in the discussion.

Figure 3 - in the analysis of sperm motility, it is worth emphasizing that it was related to total sperm motility (TMOT?).

For the reagents used in the tests (eg Trizma base, dextran), the country of origin and the name of the producer should be provided.

Literature should be carefully checked and prepared with the requirements of Animals.

Author Response

Response to Reviewer 3 Comments

Point 1. Line 72 - please provide information on how many animals were included in the study in each age group. Is n = 9 for each age group?

Response 1. Thank you for this comment. We had 9 dogs and there were 3 dogs in each age group. We collected sperm 3 times from each dog. We added this information to the material and methods.

Point 2. Line 79 - there is no information on how the sperm morphology was assessed, what dye, what method did the authors use?

Response 2. The sperm smears we stained with eosin-nigrosin (Magapor, Spain) and assessed morphology under a light microscope with a magnification of ×400.

Point 3. The analysis of the results is incomplete and needs to be completed. In the description of the methodology, the authors state that they analyzed the parameters of sperm motility using the CASA system, and therefore they should present these results (PMOT, VCL, VAP, VSL, ALH and STR values) for both extenders, taking into account different age groups.

Response 3. Thank you for this suggestion. We did not put it previously not to overload the paper. . If you think that it shout be included we will put.

Motility parameters

TMOT, %

PMOT, %

VCL, µm/s

VAP, µm/s

VSL, µm/s

ALH, µm

STR

Fresh

Group 1

84.4 ± 6.3 a

67.9 ± 6.1 a

175.3 ± 7.0a

140.5 ± 2.7a

112.8 ± 3.2 a

2.2 ± 0.3

82.2 ± 0.8

Group 2

73.3 ± 7.9 b

55.3 ± 6.1 b

171.8 ± 8.1 a

136.6 ± 3.7 a

99.7 ± 5.6 a

2.1 ± 0.2

81.5 ± 1.2

Group 3

53.9 ± 9.6c

25.7 ± 3.9d

155.8 ± 6.9 b

112.9 ±3.2 b

91 ± 5.1 b

2.0 ± 0.3

80.1 ± 0,9

Cryo with EY

Group 1

75.1 ± 5.3 b

42.1 ± 3.7c

164.3 ± 5.7a

129.8 ± 3.2 a

108.4 ± 4.1 a

2.1 ± 0.4

79.8 ± 1.1

Group 2

65.6 ± 6.2c

34.1 ± 5.0d

155.9 ± 7.2 b

110.7 ± 3.5 b

94.6 ± 3.2b

2.0 ± 0.5

78.9 ± 1.5

Group 3

33.9 ± 8.9d

18.6 ± 3.4e

148 ± 6.6b

103 ± 4.7 b

87.9 ± 3.6 b

1.9 ± 0.2

77.8 ± 0.9

Cryo with dextrann

Group 1

74.8 ± 4.5 b

40.6 ± 4.5c

166.1 ±7.2a

127.6 ± 3.2 a

106.9 ±4.9a

2.1 ± 0.2

80.1 ± 0.9

Group 2

64.2 ± 5.0c

37.0 ± 3.6d

151.2 ± 7.2 b

115.6 ± 4.3 b

92.7 ± 3.9b

1.9 ± 0.4

77.9 ±0.7

Group 3

33.3 ± 8.03d

17.8 ± 4.3e

146.7 ± 5.3 b

109 ± 3.2b

88.2 ± 3.5b

1.8 ± 0.3

79.3 ±1.1

Within a column, different superscripts indicate significant differences (p < 0.05)

Point 4. Similarly, in the analysis of sperm morphology, it would be worth showing changes within different sperm structures, i.e. head, midpiece and tails, which were assessed and listed in the methodology description (line 80).

Response 4. We did not put it previously not to overload the paper. If you think that it shout be included we will put.

Morphology abnormalities

Head

Midpiece

Tail

Multiple abnormalities

Fresh

Group 1

1.7 ±0.4a

1.8 ± 0.9 a

1.5 ±0.7 a

4.9 ±0.5 a

Group 2

3,7 ±0.9b

3.3 ±0.6 b

3.4 ±0.5b

11.11  ±1.5 b

Group 3

9.1 ±1.1c

8.9 ±0.7 a

6.3 ±0.7c

21.3 ±2.1 c

Cryo with EY

Group 1

2.3 ± 0.3 a

2.5 ± 0.5ab

3.8 ±0.4b

5.9 ±1.0 a

Group 2

3.9 ±0.7 b

4.2 ±0.3 b

7.4 ± 0.9c

10.1 ± 0.9 b

Group 3

10.0 ±1.2c

9.1 ±0.5 c

9.3 ±1.0c

22,9 ±1.9 c

Cryo with dextran

Group 1

2.1  ± 0.4 a

2.3 ± 0.4 a

3.7  ± 0.2 b

6.3  ± 0.7 a

Group 2

4.3  ± 0.4 b

4.1  ± 0.7 b

5.3  ± 0.4 b

12.2  ±1.0 b

Group 3

12,2  ±1.1 c

12.2  ± 1.2 c

11,8  ± 1.1c

17.6 ± 1.5c

Within a column, different superscripts indicate significant differences (p < 0.05)

Other comments:

Point 5. Line 86 - should be corrected ‘106 ’by ‘106’. This revision should be made throughout the text where this error occurs.

Line 76 - the sign of degrees Celsius  ‘oC’ should be corrected and elsewhere in the text.

Line 85 - should be ‘morphology’, not ‘porphology’.

The line 96 - expression ‘diluted spermatozoa’ should be replaced with e.g. ‘diluted samples of sperm’

Line 103 - expression ‘LN2’ should first provide the full name, and then use the abbreviation.

In Figure 1, there are linguistic errors that need to be corrected (it should be ‘sperm’, not ‘serm’; ‘seminal, not ‘semenal’; ‘thawing’, not ‘twaing’).

Line 141 - in the caption Table 1 is better to put ‘in the same row’ instead of ‘in the same line’.

Figure 3 and Figure 4 should be included in the results, not in the discussion.

For the reagents used in the tests (eg Trizma base, dextran), the country of origin and the name of the producer should be provided.

Literature should be carefully checked and prepared with the requirements of Animals.

Response 5. We are grateful to the Reviewer for these suggestions and corrected everything according to the recommendations.

Point 6. Figure 3 - in the analysis of sperm motility, it is worth emphasizing that it was related to total sperm motility (TMOT?).

Response 6. Thank you, we clarify the index in the text and in the title of Figure 3.

Round 2

Reviewer 2 Report

M & M

When asked why the Materials and Methods stated that VCL and VAP were measured but did not include the results, the authors responded that they did not want to burden the paper with the results, so they only included the results for total exercise volume.

However, describing content in Materials and Methods that does not describe results is nothing short of the burden on the paper.

All the contents described in the Materials and Methods section should include the results. Otherwise, the description of materials and methods should be deleted.

Author Response

Response to Reviewer 2 Comments

Point 1

When asked why the Materials and Methods stated that VCL and VAP were measured but did not include the results, the authors responded that they did not want to burden the paper with the results, so they only included the results for total exercise volume.

However, describing content in Materials and Methods that does not describe results is nothing short of the burden on the paper.

All the contents described in the Materials and Methods section should include the results. Otherwise, the description of materials and methods should be deleted.

 Response 1. Dear Reviewer, you are absolutely right. And we would have removed the extra description from the Material and Methods section; however, considering the wishes of the third reviewer to include additional indicators of motility and morphological abnormalities in the main text of the paper, we did not decrease method part and added these data. We hope you will like our paper more after new changes.

Sincerely,

Authors

Reviewer 3 Report

Dear Authors,

The manuscript has been improved. I only have one comment. I propose to include in the manuscript tables with the results of the motility parameters and morphological changes with a description.

Author Response

Response to Reviewer 3 Comments

Point 1. Dear Authors,

The manuscript has been improved. I only have one comment. I propose to include in the manuscript tables with the results of the motility parameters and morphological changes with a description.

Response 1. Dear Reviewer, thank you very much for your work. We have added the tables and their description to the main text. We hope you will like our new variant.

Sincerely,

Authors